

# Changes of collagen content in lung tissues of plateau yak and its mechanism of adaptation to hypoxia

Jingyi Li[1], Nating Huang[1], Xun Zhang[1], Ci Sun[1], Jiarui Chen[1] and Qing Wei[1,2]

[1] College of Eco-Environmental Engineering, Qinghai University, Xining, Qinghai, China
[2] State Key Laboratory of Plateau Ecology and Agriculture, Qinghai University, Xining, Qinghai, China

## ABSTRACT

Collagen is crucial for tissue structure, functional maintenance, and cellular processes such as proliferation and differentiation. However, the specific changes in collagen expression and its associated genes in the lung tissues of yaks at high altitudes and their relationship with environmental adaptation remain poorly understood. Studying differences in the content of collagen fibers and gene expression between yaks at high (4,500 m) and low (2,600 m) altitudes, as well as between cattle at low altitudes (2,600 m). Using Masson staining, we found that the collagen fiber content in the lung tissues of yaks at low altitude was significantly higher compared to yaks at high altitude and cattle at the same altitude ($P < 0.05$). It was revealed through transcriptomic analyses that genes differentially expressed between high and low altitude yaks, as well as between low altitude yaks and cattle, were notably enriched in pathways related to cell adhesion, collagen synthesis, focal adhesion, and ECM-receptor interactions. Specifically, genes involved in mesenchymal collagen synthesis (*e.g.*, COL1A1, COL1A2, COL3A1), basement membrane collagen synthesis (*e.g.*, COL4A1, COL4A2, COL4A4, COL4A6), and peripheral collagen synthesis (*e.g.*, COL5A1, COL6A1, COL6A2, COL6A3) were significantly upregulated in the lung tissues of yaks at low altitude compared to their high altitude counterparts and cattle ($P < 0.05$). In conclusion, yaks at lower altitudes exhibit increased collagen synthesis by upregulating collagen gene expression, which contributes to maintaining alveolar stability and septal flexibility. Conversely, the expression of collagen genes in yak lung tissues was down-regulated with the increase in altitude, and it was speculated that the decrease in collagen may be used to constrain the function of elastic fibers that are more abundant at high altitude, so as to enable them to adapt to the harsh environment with hypoxia and high altitude. This adaptation mechanism highlights the role of collagen in environmental acclimatization and contributes to our understanding of how altitude and species influence collagen-related physiological processes in yaks.

Corresponding author
Qing Wei, xwq3519@sina.com

## INTRODUCTION

The yak (*Bos grunniens*) is a typical plateau mammal whose biological adaptations have developed over an extended period in this environment, resulting in unique and notable characteristics (*Sapkota et al., 2022*). For example, yaks exhibit features such as efficient oxygen utilization (*Wan et al., 2021*), well-developed lung structures (*Ge et al., 2021*), adaptive hemoglobin regulation mechanisms (*Xin et al., 2019*), and specialized skin and hair that provide effective cold tolerance (*Krishnan et al., 2018*). The development and evolution of these biological traits are crucial for the survival and reproduction of yaks in highland areas. Additionally, they offer an ideal model for studying the biological mechanisms underlying animal adaptation to extreme environments.

Collagen is a crucial structural protein that serves as the fundamental unit for collagen fiber formation and is a major component of the extracellular matrix (ECM). Collagen interacts with various receptor families and cells to regulate their proliferation, migration, and differentiation. It also plays a key role in tissue connectivity, structural support, and the maintenance of physiological functions (*Gordon & Hahn, 2010*). Collagen regulation involves modulating its biosynthetic pathways through various signaling mechanisms and changes in gene expression (*Angre et al., 2022*). As of now, 28 types of collagen have been identified through molecular biology techniques and gene cloning (*Ricard-Blum, 2011*). Type I collagen is the most widely distributed, performing various functions including regulating collagen fiber formation, repairing tissue damage, and supporting and protecting the integrity of tissues and organs (*Baguisi et al., 1997*; *Saito et al., 2001*; *Chen et al., 2019*). Type I and Type III collagen are both involved in organizing collagen fibers and synthesizing collagenous materials (*Ahtikoski et al., 2001*). Type IV collagen is widely expressed in all organs and is crucial for the construction of the basement membrane in blood vessels and other tissues. It maintains membrane integrity and supports tissue-specific functions (*Pöschl et al., 2004*; *Volonghi et al., 2010*; *Mao et al., 2015*). Collagen gene expression has been shown to correlate with high-altitude acclimatization in animals (*Wiener et al., 2021*). Changes in its expression contribute to angiogenesis and vascular remodeling (*Minor & Coulombe, 2020*), regulation of alveolar surface tension (*Shi et al., 2022*), and ECM remodeling (*Zhang et al., 2011*), thereby optimizing lung structure for adaptation to hypoxic environments. Reduced collagen content in mouse lungs restricts the progression of pulmonary hypertension (*Schreier et al., 2013*; *Chen et al., 2006*). However, the relationship between changes in collagen gene expression and environmental acclimatization in the lung tissues of plateau yaks remains unknown.

Therefore, this study aimed to investigate the relationship between collagen content and gene expression changes in the lung tissues of plateau yaks and their environmental adaptation.

## MATERIALS AND METHODS

### Experimental animals

This study selected three adult male yaks from Qumalai County (QML-Y, high-altitude yak at 4,500 m), three adult male yaks from Xunhua County (XH-Y, low-altitude yak at

2,600 m), and three adult male cattle from Xunhua County (XH-C, low-altitude yellow cattle at 2,600 m), totaling nine animals. At the same elevation, XH-C served as the control group while XH-Y was the experimental group. For different elevations within the same species, XH-Y was the control group and QML-Y was the experimental group. All animals were clinically healthy. The yaks were euthanized by exsanguination in a slaughterhouse. Lungs were harvested within 30 min of death, and deep tissue samples were collected from the diaphragmatic lobe of the left lung. Some tissues were snap-frozen in liquid nitrogen for long-term storage, while smaller samples were fixed in 4% paraformaldehyde for subsequent analysis. This study was approved by the Institutional Animal Care and Use Committee (IACUC) of Qinghai University, Xining, China (Approval No.: PJ-2023037). All methods adhered to the ARRIVE 2.0 guidelines and the Guidelines for Ethical Review of Animal Welfare in Laboratory Animals (GB/T 35892-2018) of the People's Republic of China. All local regulations and laws were adhered to. All cattle and yaks used in this study were purchased from local farmers in Qinghai Province, China.

## Histologic staining

### Paraffin sections of lung tissue

Fresh lung tissue specimens from both yak and cattle were excised and immediately fixed in 4% paraformaldehyde for at least 24 h. The tissues were then trimmed to a uniform thickness of 1–2 mm and placed into designated embedding cassettes. Residual fixative was meticulously washed off with running tap water, and the tissues were air-dried before proceeding with a graded dehydration series. Dehydration began with an overnight immersion in 70% ethanol, followed by immersion in 80% ethanol the next day. The tissues were then sequentially transferred through 90% ethanol, 95% ethanol, 100% ethanol I, and 100% ethanol II for gradient dehydration. After dehydration, the specimens were cleared with xylene and embedded in paraffin. The paraffin blocks were cooled on a −20 °C freezing table, sectioned into 4 μm slices, and set aside.

## Masson staining

The paraffin sections were sequentially treated with xylene I for 8 min, xylene II for 8 min, anhydrous ethanol I for 6 min, anhydrous ethanol II for 6 min, 95% ethanol for 6 min, 85% ethanol for 6 min, and 75% ethanol for 5 min, followed by rinsing with running water. The sections were then stained with Weigert's Iron Hematoxylin Staining Solution for 5 min, washed thoroughly with water, and blued using Masson's bluing solution before another wash with water. Next, the sections were stained with Masson Lichun red acidic magenta solution for 5 min, rinsed with running water, treated with phosphomolybdic acid solution for 5 min, and then the excess liquid was removed. They were subsequently stained with aniline blue solution for 3 min, followed by rinsing with 1% glacial acetic acid aqueous solution until no blue color remained. Finally, the sections were briefly washed with 95% ethanol, dehydrated with anhydrous ethanol for 3 min, and cleared in xylene III for 5 min, xylene II for 5 min, and xylene I for 5 min before being mounted with neutral gum. Lung tissue sections were stained using the Masson method, resulting in blue-stained collagen fibers and red-stained cytoplasm. All staining solutions were prepared at identical

concentrations, and the staining times, temperatures, and washing procedures were standardized. All samples were fixed with the same fixative for an identical duration, and processed with consistent dehydration and clearing procedures. The embedding method was also uniform, ensuring that the tissue sections had the same thickness, thereby maintaining staining consistency across samples.

## Observation and measurement

Stained tissue sections were observed and photographed with a Nikon optical microscope (DS-Ri2) and subsequently measured using Image-Pro Plus 6.0 software. The measurement data were analyzed using SPSS 26.0, with results presented as mean ± standard deviation ($\bar{X}$ ± SD). A t-test was conducted to assess differences, utilizing SPSS 26.0.

## Transcriptome sequencing and analysis

Lung tissue samples were collected from cattle and yaks at the same altitude, as well as from yaks at different altitudes. RNA was extracted using the Trizol method, and mRNA was enriched with Oligo (dT) magnetic beads to construct cDNA libraries. The transcriptome was sequenced using the Illumina HiSeq 2500 platform. Raw data were filtered, and the clean reads were compared to the reference genome for gene annotation, quantitative analysis, and differential expression analysis. In the gene analysis section, we also performed analyses for gene coverage, sequencing randomness, and sequencing saturation. Gene coverage represents the percentage of bases in a gene that are covered by reads, calculated as the ratio of the bases covered by reads to the total bases in the gene's coding region. To evaluate the randomness of mRNA interruption, we analyzed the distribution of reads across reference genes. Due to variations in the lengths of different reference genes, we normalized read positions to relative positions (the ratio of read position to gene length * 100) and counted reads across different positional ratios. Ideally, with good randomness, reads should be evenly distributed across gene sites. Sequencing saturation analysis assesses whether the sequencing depth of a sample is sufficient to detect most genes. As the sequencing depth (number of reads) increases, the number of detected genes rises until reaching a plateau, indicating saturation in gene detection. The input data for gene differential expression analysis consisted of read count data from gene expression level assessments. The differential expression analysis was performed using DESeq2 and was divided into three main steps: normalization of read counts, calculation of $P$-values based on the model, and correction for multiple hypothesis testing to obtain the false discovery rate (FDR). Genes with |log2(FoldChange)| > 0.585 and FDR < 0.05 were identified as significant, based on variance analysis results. This set of significant genes was then clustered and analyzed.

To ensure the reliability and reproducibility of the experimental results, this study employed both biological and technical replicates. For biological replication, nine independent samples were collected from local herders and divided into three groups, with each group undergoing three replicates under identical conditions. Technical replication involved conducting all experiments under stringent quality control, maintaining
**Table 1 Gene primer sequence information.**

| Gene name | Primer types | Sequence (5′-3′) |
|---|---|---|
| COL1A1 | Upstream primers | TCCTTCTGGTCCTCGTGGTCTC |
| | Downstream primers | TCGCCATCATCTCCGTTCTT |
| COL1A2 | Upstream primers | GCCACCCAGAATGGAGCAGT |
| | Downstream primers | GATGCAGGTTTCGCCAGTAGA |
| COL6A2 | Upstream primers | CGGGTGCTGTGACTGTGAGAA |
| | Downstream primers | CAGCCTGTTGACCACGTTGAT |
| GAPDH | Upstream primers | GGTCACCAGGGCTGCTTTTA |
| | Downstream primers | CCAGCATCACCCCACTTGAT |

consistent temperature, time, and reagent lot number. The statistical power of this experimental design, calculated in RNASeqPower is 0.95 (https://rodrigo-arcoverde. shinyapps.io/rnaseq_power_calc/). This result illustrates the validity and rigor of our transcriptome sequencing.

## Fluorescence quantitative PCR

Real-time quantitative PCR was performed using cDNAs from lung tissues of yak and cattle collected from high and low altitudes. Relative mRNA levels were measured using 7.5 µl of 2× Universal Blue SYBR Green qPCR Master Mix, 1.5 µl of 2.5 µM gene primers (both upstream and downstream), 2.0 µl of the reverse transcription product (cDNA), and 4.0 µl of nuclease-free water. *GAPDH* served as the internal reference gene, with the primer sequences listed in Table 1. The mRNA expression of *COL1A1*, *COL1A2*, and *COL6A2* genes in lung tissues was analyzed using the $2^{-\Delta\Delta CT}$ method for relative quantification. An analysis of variance was conducted using SPSS 26.0 software to identify significant differences.

## RESULTS

### Masson staining results of yak and cattle lung tissues

The blue-stained regions of the lung tissue sections revealed collagen fiber structures. Collagen fibers were more abundant in the lung tissues of yaks at low altitude compared to both cattle at low altitude and yaks at high altitude (Fig. 1A). Quantitative analysis revealed that the percentage of collagen fibers in the lung tissues of yaks at low altitude was significantly higher than in cattle ($P = 0.000$) (Fig. 1B). Additionally, the percentage of collagen fibers in the lung tissues of yaks at high altitude was significantly lower than in yaks at low altitude ($P = 0.008$) (Fig. 1C).

### Overall statistics of differential genes between groups

The results revealed 3,684 differentially expressed genes in yaks compared to cattle in low-altitude areas, including 2,189 up-regulated genes and 1,495 down-regulated genes (Fig. 2A). Additionally, there were 2,326 differentially expressed genes when comparing

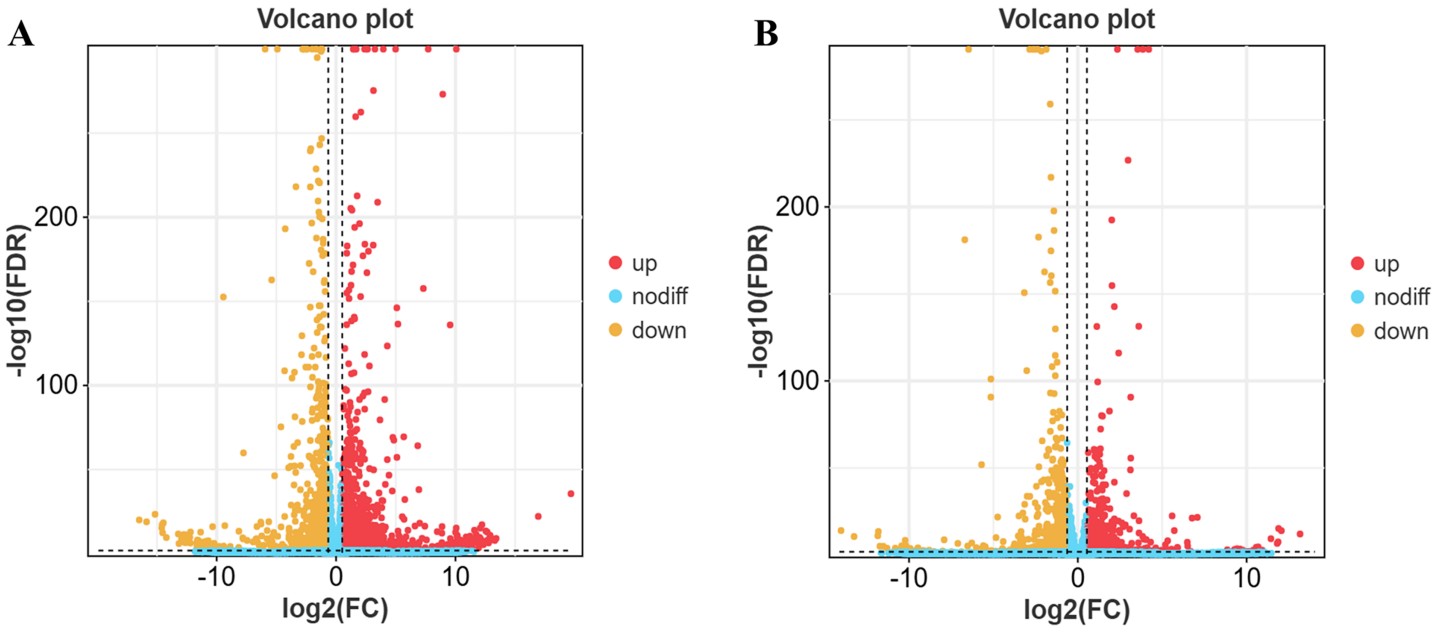

**Figure 1 Masson staining of yak and cattle lung tissue.** (A) Masson staining of XH-C, XH-Y and QML-Y. (B) Percentage of collagen fibers in XH-Y *vs* XH-C lung tissues. (C) Percentage of collagen fibers in QML-Y *vs* XH-Y lung tissues. Asterisks (**) denote a highly significant difference, $P < 0.01$.

**Figure 2 Differential gene volcano map.** (A) XH-Y *vs* XH-C. (B) QML-Y *vs* XH-Y.

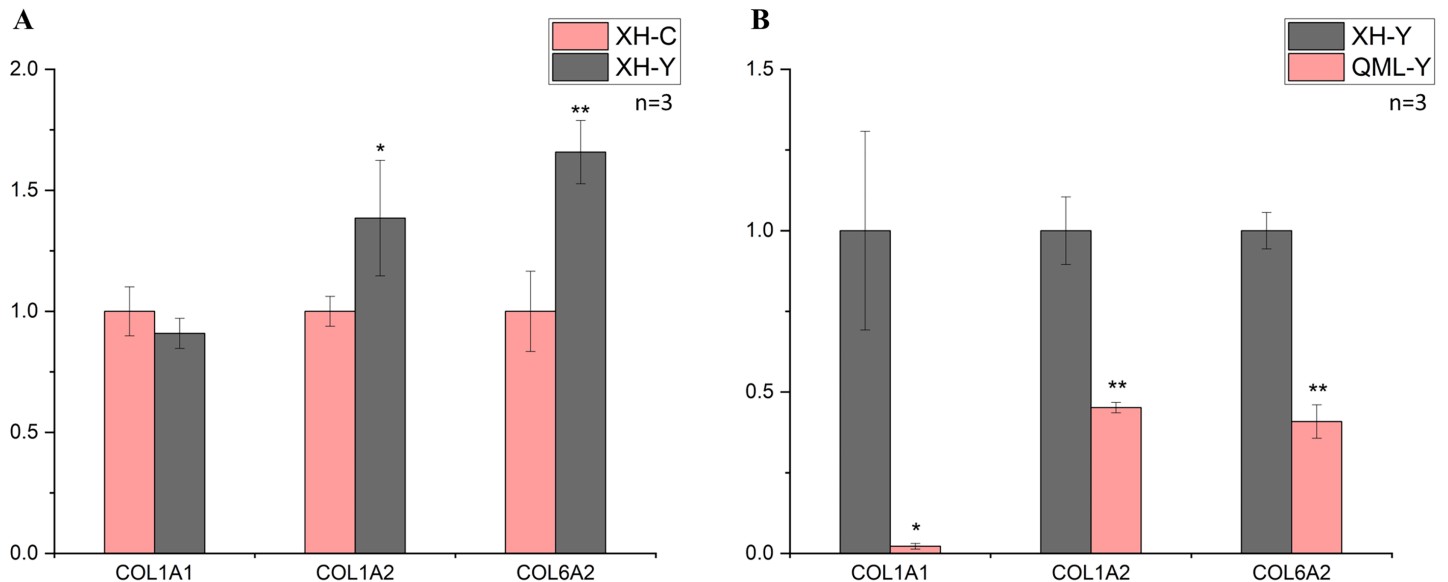

**Figure 3 Changes in mRNA expression levels of collagen-related genes.** (A) XH-Y *vs* XH-C. (B) QML-Y *vs* XH-Y. An asterisk (*) denotes significant difference, *P* < 0.05. Two asterisks (**) denote highly significant difference, *P* < 0.01.

yaks from high and low-altitude areas, with 1,340 up-regulated genes and 986 down-regulated genes (Fig. 2B).

## Fluorescence quantitative PCRvalidation analysis

In the low-altitude comparison between yaks and cattle, relative quantitative analysis was performed on the *COL1A1*, *COL1A2* and *COL6A2* genes using cattle as the standard (Fig. 3A). The *COL1A1* gene showed no significant differential expression between the two groups (*P* = 0.247), while the *COL1A2* gene exhibited significant differential expression (*P* = 0.049), and the *COL6A2* gene displayed highly significant differential expression (*P* = 0.006).

In the comparison between high-altitude and low-altitude yaks, the *COL1A1*, *COL1A2* and *COL6A2* genes were analyzed using relative quantitative methods with low-altitude yaks as the reference (Fig. 3B). The *COL1A1* gene showed significant differential expression between the two groups (*P* = 0.029), while the *COL1A2* (*P* = 0.001) and *COL6A2* (*P* = 0.000) genes exhibited highly significant differential expression. The quantitative detection results were consistent with the transcriptome sequencing trends.

## Gene ontology enrichment analysis

Gene Ontology (GO) analysis revealed that low-altitude yaks were enriched in biological processes such as bioadhesion, cell adhesion, and extracellular matrix organization, as well as cellular components including cell periphery, extracellular matrix, and collagen-containing extracellular matrix, compared to low-altitude cattle (Fig. 4A); and high-altitude yaks were enriched in similar biological processes and cellular components compared to low-altitude yaks (Fig. 4B). These findings indicate that differentially expressed genes in lung tissues are involved in cell adhesion and collagen synthesis in both groups.

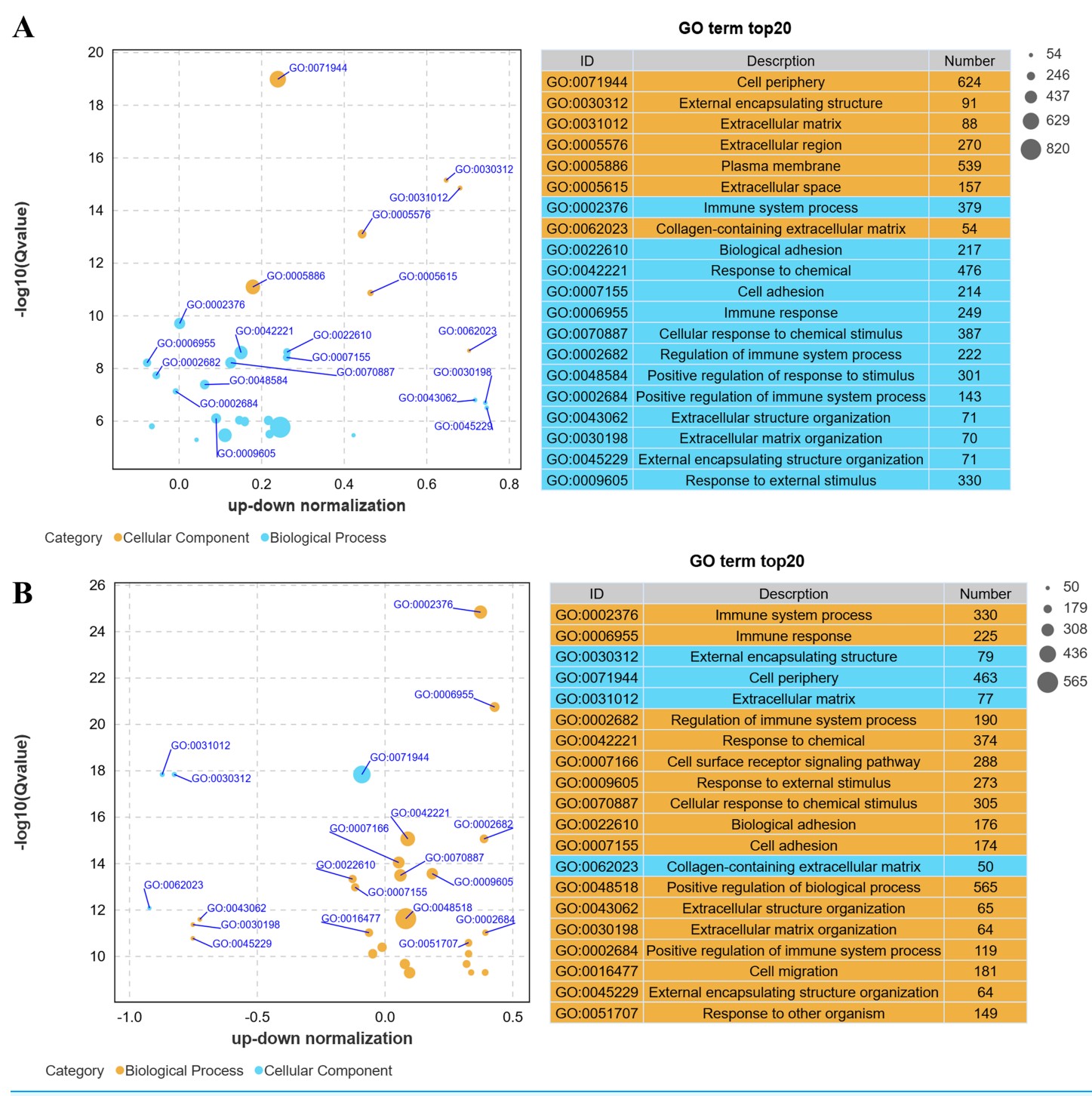

**Figure 4  Bubble plot of GO enrichment differences.** (A) XH-Y *vs* XH-C. (B) QML-Y *vs* XH-Y.  

## Kyoto encyclopedia of genes and genomes enrichment analysis

Differentially expressed genes in the lung tissues of yaks and cattle at low altitude were annotated using the Kyoto Encyclopedia of Genes and Genomes (KEGG) database. Analysis of the top 20 pathways with the most significant differences (smallest Q-value)

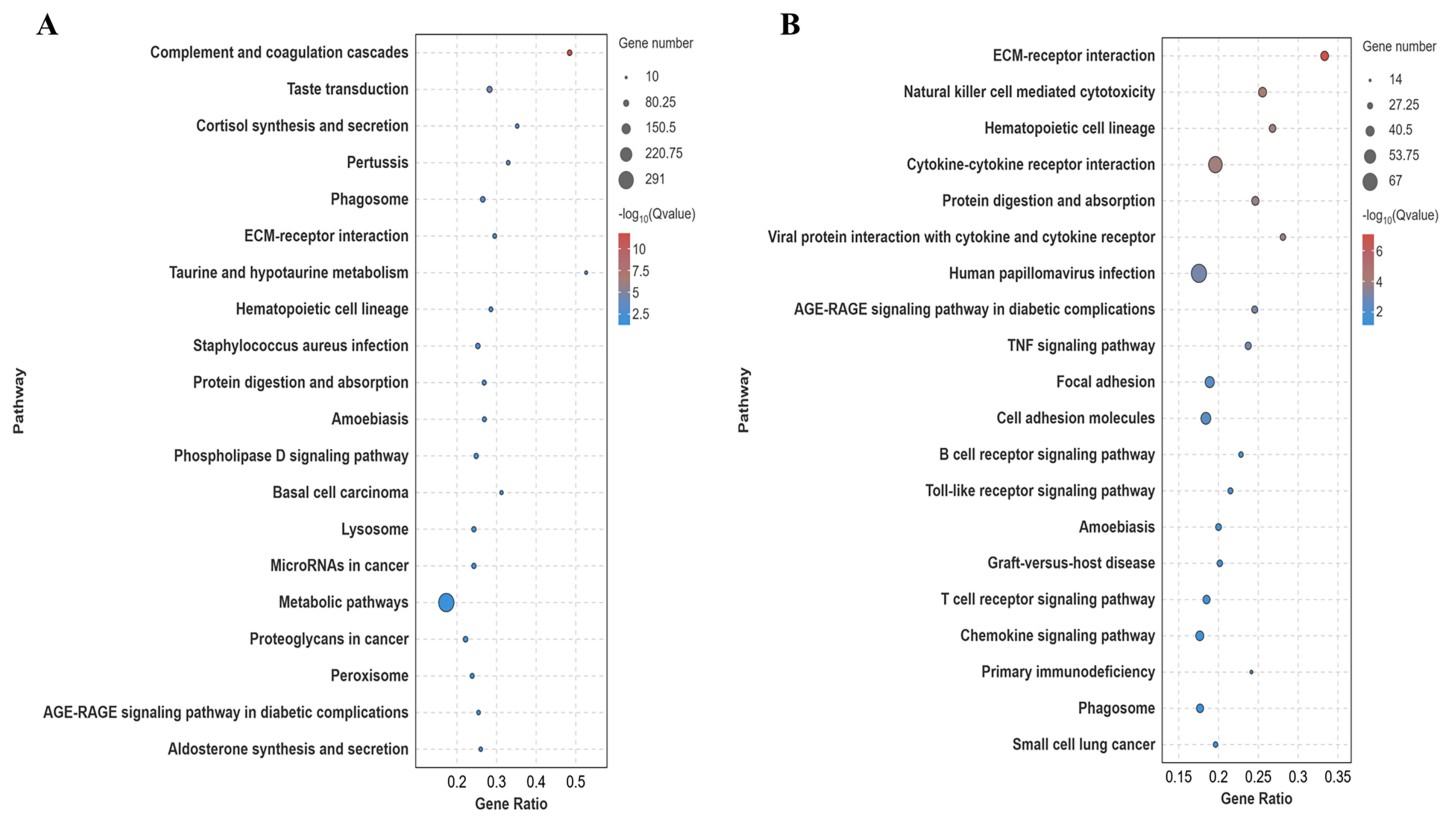

**Figure 5 KEGG significance bubble plot.** (A) XH-Y *vs* XH-C. (B) QML-Y *vs* XH-Y.

revealed that these genes were enriched in signaling pathways such as ECM-receptor interaction and protein digestion and absorption (Fig. 5A). Similarly, differentially expressed genes in yak lung tissues from both high and low altitude regions were annotated using the KEGG database. Analysis of the top 20 pathways with the most significant differences (smallest Q-value) showed enrichment in signaling pathways including ECM-receptor interaction, protein digestion and uptake, focal adhesion, and cell adhesion molecules (Fig. 5B). These results suggest that the differentially expressed genes in the lung tissues of yaks and cattle at low altitude, as well as yaks at both high and low altitudes, are enriched in pathways related to collagen synthesis.

## Screening of differentially expressed collagen genes

Based on existing studies and GO annotation, 19 significantly different collagen genes were identified in the lung tissues of yak and cattle at low altitude. Among these, 18 genes were up-regulated and one gene was down-regulated in yak lung tissues (Table 2). Additionally, 23 significantly different collagen genes were found in yak lung tissues at both high and low altitudes, with all of these genes being down-regulated at high altitude (Table 3).

## Significant differential gene expression changes in collagen

The differentially expressed collagen genes in the two groups were categorized into three types: mesenchymal collagen synthesis-related genes, basement membrane collagen

**Table 2 Annotation of XH-Y *vs* XH-C significantly different collagen-related genes.**

| ID | Symbol | Description |
|---|---|---|
| ENSBGRG00000005548 | COL1A1 | XP_006041214.2 collagen alpha-1 (I) chain isoform X1 |
| ENSBGRG00000013466 | COL1A2 | Collagen type I alpha 2 chain |
| ENSBGRG00000003247 | COL3A1 | Collagen type III alpha 1 chain |
| ENSBGRG00000004437 | COL4A1 | Collagen type IV alpha 1 chain |
| ENSBGRG00000005083 | COL4A2 | Collagen type IV alpha 2 chain |
| ENSBGRG00000012460 | COL4A4 | Collagen type IV alpha 4 chain |
| ENSBGRG00000021026 | COL4A6 | Collagen type IV alpha 6 chain |
| ENSBGRG00000005255 | COL5A1 | Collagen type V alpha 1 chain |
| ENSBGRG00000000813 | COL6A1 | Collagen type VI alpha 1 chain |
| ENSBGRG00000000863 | COL6A2 | Collagen type VI alpha 2 chain |
| ENSBGRG00000017439 | COL6A3 | Collagen alpha-3(VI) chain-like |
| ENSBGRG00000003956 | COL6A5 | Collagen type VI alpha 5 chain |
| ENSBGRG00000016164 | COL14A1 | Collagen type XIV alpha 1 chain |
| ENSBGRG00000020183 | COL15A1 | Collagen type XV alpha 1 chain |
| ENSBGRG00000008612 | COL16A1 | Collagen type XVI alpha 1 chain |
| ENSBGRG00000024606 | COL21A1 | Collagen type XXI alpha 1 chain |
| ENSBGRG00000011762 | COL23A1 | Collagen type XXIII alpha 1 chain |
| ENSBGRG00000025649 | COL26A1 | Collagen type XXVI alpha 1 chain |
| ENSBGRG00000007346 | COL28A1 | Collagen type XXVIII alpha 1 chain |

**Table 3 Annotation of QML-Y *vs* XH-Y significantly different collagen-related genes.**

| ID | Symbol | Description |
|---|---|---|
| ENSBGRG00000005548 | COL1A1 | XP_006041214.2 collagen alpha-1(I) chain isoform X1 |
| ENSBGRG00000013466 | COL1A2 | Collagen type I alpha 2 chain |
| ENSBGRG00000024422 | COL2A1 | Collagen type II alpha 1 chain |
| ENSBGRG00000003247 | COL3A1 | Collagen type III alpha 1 chain |
| ENSBGRG00000004437 | COL4A1 | Collagen type IV alpha 1 chain |
| ENSBGRG00000005083 | COL4A2 | Collagen type IV alpha 2 chain |
| ENSBGRG00000012460 | COL4A4 | Collagen type IV alpha 4 chain |
| ENSBGRG00000021853 | COL4A5 | Collagen type IV alpha 5 chain |
| ENSBGRG00000021026 | COL4A6 | Collagen type IV alpha 6 chain |
| ENSBGRG00000005255 | COL5A1 | Collagen type V alpha 1 chain |
| ENSBGRG00000003463 | COL5A2 | Collagen type V alpha 2 chain |
| ENSBGRG00000001716 | COL5A3 | Collagen type V alpha 3 chain |
| ENSBGRG00000000813 | COL6A1 | Collagen type VI alpha 1 chain |
| ENSBGRG00000000863 | COL6A2 | Collagen type VI alpha 2 chain |
| ENSBGRG00000017439 | COL6A3 | Collagen alpha-3(VI) chain-like |
| ENSBGRG00000002584 | COL8A1 | Collagen type VIII alpha 1 chain |

| ID | Symbol | Description |
| --- | --- | --- |
| ENSBGRG00000017451 | COL9A2 | Collagen type IX alpha 2 chain |
| ENSBGRG00000006189 | COL12A1 | Collagen type XII alpha 1 chain |
| ENSBGRG00000020183 | COL15A1 | Collagen type XV alpha 1 chain |
| ENSBGRG00000008612 | COL16A1 | Collagen type XVI alpha 1 chain |
| MSTRG.1779 | COL18A1 | XP_024830811.1 collagen alpha-1(XVIII) chain isoform X2 [Bos taurus] |
| ENSBGRG00000024606 | COL21A1 | Collagen type XXI alpha 1 chain |
| ENSBGRG00000025649 | COL26A1 | Collagen type XXVI alpha 1 chain |

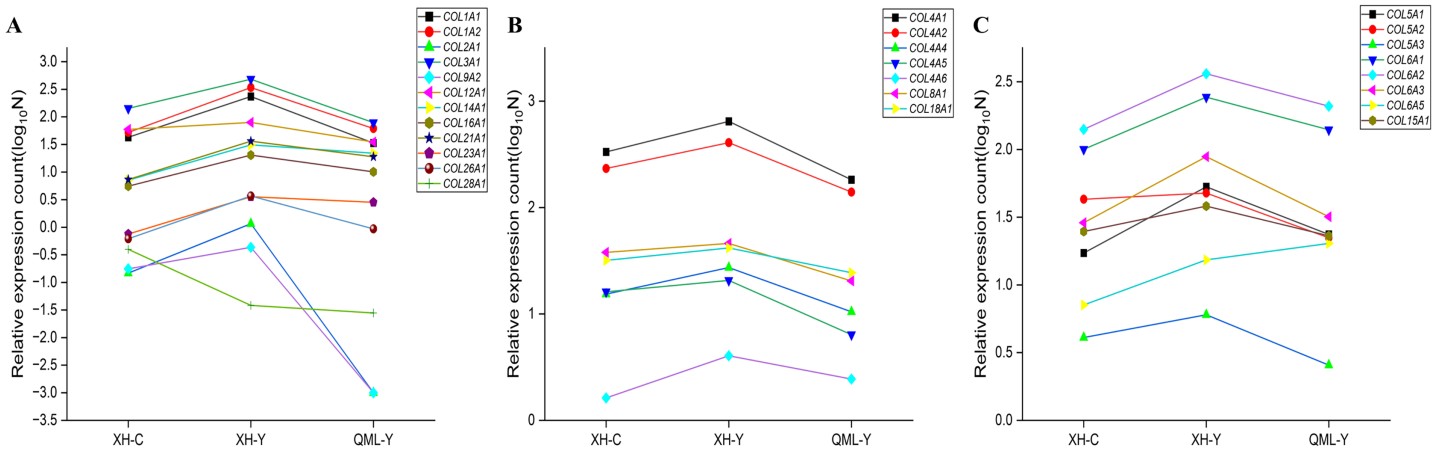

**Figure 6 Significant differential gene expression level changes in collagen.** (A) Genes related to mesenchymal collagen synthesis. (B) Genes related to basement membrane collagen synthesis. (C) Genes related to peripheral cellular collagen synthesis.

synthesis-related genes, and peripheral cellular collagen synthesis-related genes. Specific values of relative gene expression are presented in Supplemental Material 1. Among these, interstitial collagen synthesis-related genes such as *COL1A1*, *COL1A2*, *COL2A1*, *COL3A1*, *COL9A2*, *COL12A1*, *COL14A1*, *COL16A1*, *COL21A1*, *COL23A1*, and *COL26A1* were highly expressed in the lung tissues of yak from low-altitude areas compared to both cattle from low-altitude areas and yak from high-altitude areas (Fig. 6A). Basement membrane collagen synthesis-related genes, including *COL4A1*, *COL4A2*, *COL4A4*, *COL4A5*, *COL4A6*, *COL8A1*, and *COL18A1* were also highly expressed in the lung tissues of yak from low-altitude areas compared to cattle from low-altitude areas and yak from high-altitude areas (Fig. 6B). Peripheral collagen synthesis-related genes such as *COL5A1*, *COL5A2*, *COL5A3*, *COL6A1*, *COL6A2*, *COL6A3*, and *COL15A1* were highly expressed in the lung tissues of yak from low-altitude areas compared to cattle from low-altitude areas and yak from high-altitude areas. In contrast, *COL6A5* was more highly expressed in the lung tissues of yak from high-altitude areas compared to those from low-altitude areas and cattle (Fig. 6C).

## DISCUSSION

Lung tissue is the primary organ for oxygen entry into the body, with the alveoli serving as the main site for gas exchange and the functional unit of lung tissue (*Warburton, 2021*). As a hypoxia-sensitive organ, the structure of lung tissue varies among species at the same altitude and within the same species at different altitudes to meet the organism's oxygen demands (*Jing et al., 2022*). GO functional analysis indicated that differentially expressed genes in the lung tissues of yaks at high and low altitudes, as well as between cattle at low altitudes, are involved in processes like cell adhesion and collagen synthesis. KEGG analysis further demonstrated that differentially expressed genes in lung tissues of yaks and cattle at low altitude, as well as yaks at high and low altitudes, were enriched in pathways related to adhesion (*e.g.*, focal adhesion and cell adhesion molecules) and collagen synthesis (*e.g.*, ECM-receptor interactions and protein digestion and uptake). The extracellular matrix (ECM), composed of the basement membrane (BM) and intercellular matrix, encompasses structures like collagen and elastic fibers (*Kular, Basu & Sharma, 2014*). The ECM plays a crucial role in tissue and organ morphogenesis, and in maintaining cellular and tissue structure and function (*Krieg & Aumailley, 2011*). Cellular adhesion serves as a key mechanism for cells to connect to the ECM, with collagen playing a pivotal role in this process (*Mishra & Manavathi, 2021*; *Yamaguchi & Knaut, 2022*). Based on this, we hypothesize that plateau yaks regulate collagen and elastin synthesis, thereby influencing the cell adhesion process. This, in turn, regulates lung tissue function.

*Zhou et al. (2015)* observed that small pulmonary arterioles in the lungs of adult yellow cattle contained fewer elastic fibers compared to similar-sized arterioles in adult yaks. Elastic fibers confer elasticity to vessel walls, while collagen fibers provide tensile strength (*Zulliger et al., 2004*; *Kumar et al., 2013*). Additionally, elastic fibers help maintain collagen fiber conformation, and their combination enhances the resistance of the vessel walls to compression (*Mizuno et al., 2005*). Research has shown that alveolar septa in plateau yaks are significantly thicker than those in plains cattle (*Zhang et al., 2023*). The presence of collagen fibers in these septa contributes to their flexibility, which is crucial for lung tissue function (*Mascaretti et al., 2009*). Meanwhile, the present study found that collagen fibers were more abundant in the lung tissue of yaks at low altitudes compared to cattle, and the expression of collagen synthesis-related genes was also higher in yaks. Among these genes, *COL3A1*, which is involved in collagen synthesis in mesenchymal tissues, is crucial for maintaining the morphology and structure of the skin, tissues, and organs. Up-regulation of *COL3A1* in the dense layer of sheep fetal membranes has been shown to enhance membrane tensile strength (*Mi et al., 2018*). Additionally, *COL3A1* facilitates cell adhesion, migration, proliferation, and differentiation through interactions with integrins, which are cell surface receptors (*Kim et al., 2005*; *Kuivaniemi & Tromp, 2019*). Additionally, the peripheral collagen synthesis-related genes *COL6A1* and *COL6A2* are crucial for supporting tissue structure, maintaining extracellular matrix elasticity, and participating in cellular signaling (*Fitzgerald et al., 2008*). They also interact with other collagens and basement membranes to reinforce the extracellular matrix network (*Bonaldo et al., 1990*). The interstitium supports the alveolar network through collagen, regulates intercellular

adhesion, and is critical for maintaining lung tissue integrity (*Baldock et al., 2003*; *Atkinson et al., 2008*; *Voiles et al., 2014*). Yak lung tissue, which is rich in collagen fibers, exhibits greater resistance to compression and enhanced flexibility compared to cattle lung tissue. Additionally, the basement membrane of yak lung tissue is more structurally stable and functionally superior to that of cattle, featuring increased extracellular matrix elasticity, a more extensive extracellular matrix network, and stronger interactions with other molecules.

Studies have demonstrated that lung tissues of high-altitude indigenous animals, including Tibetan sheep, plateau pika, plateau zokor, and Himalayan dry otter, contain more elastin fibers (*Zhao et al., 2022*; *West et al., 2021*). Specifically, yak lungs at high altitudes have a higher concentration of elastin fibers compared to those at lower altitudes, which helps ensure adequate ventilation (*Li et al., 2021*). However, excessive elastin deposition can impair ventilation and gas exchange, leading to reduced lung compliance (*Yombo et al., 2023*). In a healthy lung, a balance between collagen and elastin fibers is maintained, providing both structural strength to resist pressure and sufficient elasticity to accommodate respiratory movements (*Halsey et al., 2023*). Conversely, collagen fibers are highly resistant to tensile forces and may constrain the elastic function of elastin fibers (*Kim et al., 2015*; *Rabkin, 2017*). Abnormal deposition of collagen and extracellular matrix (ECM) components in lung tissue is a hallmark of pulmonary fibrosis (*Quesnel et al., 2010*). *Chen et al. (2023)* also found that Collagen I and Collagen III proteins were significantly increased in yak lung fibrotic tissues, indicating that collagen-related genes play a crucial role in the fibrosis process in yak lung tissues. In the present study, we observed that collagen fibers were more abundant in the lung tissues of yaks at low altitude compared to those at high altitude. Additionally, the expression of collagen synthesis-related genes was higher in the lung tissues of yaks at low altitude. Notably, the down-regulation of *COL1A1*, a gene involved in interstitial collagen synthesis, exhibited an anti-fibrotic effect (*Niu et al., 2022*). *Gong et al. (2023)* found that inhibiting *COL1A1* expression reduced the severity of pulmonary fibrosis induced in mice. Basement membrane collagen synthesis-related genes primarily include type IV collagen, such as *COL4A1*, *COL4A2*, *COL4A4*, and *COL4A6* (*Yurchenco, 2011*), with *COL4A1* also being closely linked to cell proliferation (*Jin et al., 2017*). Peripheral collagen synthesis genes, such as *COL5A1* and *COL5A2*, are often found alongside other collagen types. These genes contribute to the structural framework of tissues, extracellular matrix assembly, and cell signaling (*Koch et al., 2001*). Alveolar hypoxia, prevalent in non-high-altitude-adapted species, can contribute to the onset of pulmonary diseases such as pulmonary hypertension and fibrosis (*Sydykov et al., 2021*; *Schoene, 1999*). Pulmonary artery compliance decreases while collagen deposition increases in the context of pulmonary hypertension (*Thenappan et al., 2016*). Reduced collagen accumulation in mice has been shown to aid in preserving near-normal compliance and vascular resistance properties of the pulmonary circulation under hypoxic conditions, thereby mitigating the onset of pulmonary hypertension (*Chen et al., 2006*). Pulmonary fibrosis can be significantly improved through the inhibition of myofibroblast differentiation and collagen accumulation in rats

(*Andugulapati et al., 2020*). Therefore, reducing collagen fiber content in yak lung tissues at high altitudes may help mitigate the excessive accumulation of fibers due to elevated elastic fiber content. Lower expression of interstitial collagen synthesis-related genes helps reduce excessive fiber accumulation in lung tissues caused by the low-oxygen plateau environment, thus better protecting yak lung tissues and aiding adaptation to the harsh plateau conditions. The small sample size due to logistical challenges, so only the available samples were comparatively analyzed in relation to other animals in hypoxic environments. Future work will involve a comprehensive analysis of different fiber types in lung tissues to elucidate their relationships and to explain how structural and compositional changes in fibers impact the adaptation of plateau yaks to their environment.

## CONCLUSIONS

Differentially expressed genes in the lung tissues of yak and cattle at low altitude, as well as yak at both high and low altitudes, are enriched in pathways and functions related to collagen synthesis. At the same altitude, yaks can increase collagen content by upregulating collagen-related genes, compared to cattle. This upregulation helps maintain the stability of alveolar structures, the flexibility of alveolar septa, and the integrity of the basement membrane in yaks. As altitude increases, collagen genes may be downregulated to decrease collagen content in yak lung tissues. This reduction helps limit the elasticity of elastic fibers, mitigates the fibrotic response in lung tissues, and aids in adaptation to the harsh hypoxic environment.

### Funding

This work was supported by the Qinghai Basic Research Project (grant number: 2023-ZJ-708). The funders had no role in study design, data collection and analysis, decision to publish, or preparation of the manuscript.

### Grant Disclosures

The following grant information was disclosed by the authors:
Qinghai Basic Research Project: 2023-ZJ-708.

### Competing Interests

The authors declare that they have no competing interests.

### Author Contributions

- Jingyi Li conceived and designed the experiments, performed the experiments, analyzed the data, prepared figures and/or tables, authored or reviewed drafts of the article, and approved the final draft.
- Nating Huang performed the experiments, prepared figures and/or tables, and approved the final draft.

- Xun Zhang analyzed the data, prepared figures and/or tables, and approved the final draft.
- Ci Sun analyzed the data, prepared figures and/or tables, and approved the final draft.
- Jiarui Chen conceived and designed the experiments, authored or reviewed drafts of the article, and approved the final draft.
- Qing Wei conceived and designed the experiments, authored or reviewed drafts of the article, and approved the final draft.

## Animal Ethics

The following information was supplied relating to ethical approvals (*i.e.*, approving body and any reference numbers):

This study was approved by the Institutional Animal Care and Use Committee of Qinghai University (Xining, China).

## Data Availability

The sequence data is available at NCBI SRA: PRJNA1127767.

## Supplemental Information

Supplemental information for this article can be found online at http://dx.doi.org/10.7717/peerj.18250#supplemental-information.

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
