# Peer review of "Changes of collagen content in lung tissues of plateau yak and its mechanism of adaptation to hypoxia"

_PeerJ, doi:10.7717/peerj.18250_

## Round 0.1 · original submission · Major Revisions

Dear authors, The manuscript contains a lot of controversial data, which have been analyzed quite thoroughly by reviewer 2. If you respond very carefully to all the comments in both reviews, the article can probably be published. It is probably necessary to add data (the sample is very small for a similar study). I am not sure that you will be able to collect new data quickly enough. However, I believe it is necessary to give you the opportunity to improve this manuscript as much as possible so that the reviewers will have the opportunity to approve its publication when they review it again.

Reviewer 1 ·

Basic reporting

Changes of collagen content in lung tissues of plateau yak and a preliminary study of its mechanism (preliminary study of its mechanism – what exactly did you mean by that?)
The goal needs to be aligned with the title of the article.
In Figures 1 and 3, n =3 must be indicated. In Figure 3, the authors present the significance above the control group column. This must be revised, placing the asterisks above the experimental group column XH-Y.
It is preferable to make the figures clear.
Move the text "Differences in gene expression between the two groups were analyzed using differential expression analysis software (DESeq2), and the test parameters |log2(FoldChange)| > 0.585 and FDR < 0.05" to the "Materials and Methods" section.
Subsection titles should be written without abbreviations. If abbreviations are used, they should be defined first. For example, "GO enrichment analysis." After the title, the authors should specify that GO stands for Gene Ontology.
The authors mention mouse kidneys in the discussion, although the paper focuses on lung tissue. In this section, it is necessary to compare your research with other authors explicitly related to the topic.
The manuscript should be checked by a fluent English speaker

Experimental design

The authors present two control and two experimental groups in the materials and methods section. One group serves as both the control and the experimental group. For better clarity, presenting them as Group I, Group II, and Group III would be preferable.

Validity of the findings

In this section, prospects for future research can also be presented

Reviewer 2 ·

Basic reporting

The manuscript requires several improvements to meet basic reporting standards:
The introduction lacks a clear and focused hypothesis or research question. The connection between collagen expression in yak lungs and high-altitude adaptation is not well-established. While the introduction references relevant literature, it fails to critically evaluate these sources in the context of the study. A more focused discussion on how previous research directly relates to the study's objectives would provide better context for the reader.
The use of only three animals per group raises concerns about the statistical power and robustness of the findings. Increasing the sample size could provide more reliable results.
The exact timing and conditions under which the lung tissue samples were collected post-mortem are not clearly described. This is crucial as post-mortem changes can affect gene expression and tissue integrity.
While standard procedures are followed, there is insufficient detail on the quality control measures during histological staining. More information on how staining consistency was ensured across samples would improve transparency.
The results section lacks clarity in data presentation. For instance, specific results related to collagen expression levels across different groups are not adequately detailed, making it difficult to assess the findings. Including more descriptive statistics and clearer visual aids (figures and tables) would enhance comprehension.
The discussion of results is somewhat superficial and does not thoroughly explore the biological significance of the findings. There is a need for deeper analysis and interpretation of how the observed changes in collagen expression relate to high-altitude adaptation.
The discussion does not effectively link the results back to the initial hypothesis regarding collagen's role in high-altitude adaptation in yaks. Strengthening this connection would improve the coherence of the manuscript.
There is limited comparison with existing literature on collagen expression and environmental adaptation. A more comprehensive discussion that situates the findings within the broader context of animal adaptation to extreme environments would provide more value to the reader.
Some parts of the discussion appear speculative without sufficient evidence or support from the data presented. These should be either backed by data or rephrased to reflect uncertainty.

Experimental design

The sample size of nine animals (three in each group) is quite small, potentially limiting the statistical power of the study and making it difficult to draw robust conclusions. This small sample size may not capture the natural variability within and between groups.
The selection of animals from different altitudes and species introduces multiple variables (altitude, species, individual variation) that may confound the results, especially given the small sample size. There is a risk that the study may not adequately control for these confounding variables.
The study lacks a clear rationale for the specific choice of altitudes (2600 m and 4500 m) and how these specific altitudes relate to the physiological differences being studied. A broader range of altitudes or a more detailed justification for the chosen levels could strengthen the study's conclusions.
The method of sacrificing animals by exsanguination and immediate tissue collection may introduce variability in the quality of the lung samples, as differences in the time between death and tissue preservation could affect RNA integrity and histological appearance.
The study mentions adherence to ethical guidelines, but there is no detailed description of the procedures used to minimize animal suffering, which could be an important consideration for ethical transparency.
The histological methods used, such as paraffin embedding and Masson staining, are standard, but the description of the protocol lacks details about potential sources of variability, such as the exact times and temperatures used during processing and staining. These factors could affect the quality and consistency of the histological results.
The use of a single reference gene (GAPDH) for normalization in qPCR experiments is a potential flaw, as it assumes that this gene is stably expressed across all samples and conditions. Multiple reference genes are usually recommended to ensure accurate normalization.
The RNA sequencing methodology is not fully described in terms of how sequencing depth, coverage, and other quality control metrics were determined and validated. Insufficient sequencing depth could lead to incomplete or biased gene expression profiles.
The statistical analysis mentions the use of ANOVA and a calculated statistical power, but there is no discussion of how potential batch effects or other sources of technical variability were addressed. This could be particularly important given the small sample size.
There is no mention of how the authors handled multiple comparisons in the RNA sequencing data, which is critical given the large number of genes analyzed. Without correction for multiple testing, there is a risk of false-positive results.
The methodology for selecting and validating differentially expressed genes from RNA sequencing data is not fully detailed. Clear criteria for significance and validation of key findings through independent methods (e.g., additional qPCR) are necessary to confirm the reliability of the results.

Validity of the findings

Discussed in the previous section

Additional comments

The english language of the whole text must be improved
Some comments about discussion section:
The discussion section presents somewhat contradictory conclusions regarding the relationship between collagen and elastic fibers in yak lung tissues at different altitudes. On one hand, it suggests that increased collagen content contributes to structural stability and flexibility, particularly at low altitudes. On the other hand, it argues that a reduction in collagen content at high altitudes may be adaptive to avoid excessive fiber deposition and fibrosis. The explanation lacks clarity and fails to convincingly reconcile these opposing interpretations.
The discussion tends to overinterpret the data, particularly when hypothesizing about the functional implications of gene expression changes. For example, while the study found differences in collagen-related gene expression, the direct link to functional outcomes like lung tissue elasticity, compression resistance, and fibrosis is speculative and not fully supported by the presented data. These inferences would require additional functional assays or in vivo studies to be validated.
Although the discussion mentions some studies on collagen and elastic fibers in other species, it lacks a thorough integration with the broader body of literature on lung adaptation to high altitude. The discussion could benefit from a more comprehensive comparison with findings from other high-altitude animals and a critical evaluation of whether the observed changes in yak are unique or part of a broader adaptive strategy seen across species.
The discussion does not adequately explore alternative explanations for the observed differences in collagen content and gene expression. For instance, environmental factors other than altitude, such as diet, temperature, or physical activity, could also influence collagen expression. Considering these factors could provide a more nuanced interpretation of the results.
The discussion does not address the limitations related to the small sample size and its potential impact on the generalizability of the findings. Acknowledging these limitations is crucial for accurately interpreting the results and for proposing future studies to confirm the hypotheses generated.

---

## Round 0.2 · Minor Revisions

Authors should improve the manuscript according to the few remaining comments from reviewers.

Reviewer 1 ·

Basic reporting

no comment

Experimental design

no comment

Validity of the findings

no comment

Additional comments

A number of sentences need to be revised in the article's abstract.
This study aims to address this gap by examining the differences in collagen fiber content and gene expression between yaks at high altitude (4500 m) and those at low altitude (2600 m), as well as between yaks and cattle at the same altitude
Change for
studying differences in the content of collagen fibers and gene expression between yaks at high (4500 m) and low (2600 m) altitudes, as well as between cattle at low altitudes (2600 m).

t is not desirable to write in the abstract «we revealed». Instead should be written «it was revealed»

The authors cite two goals. You only need to submit one
1. Therefore, this study aimed to investigate the relationship between collagen content and gene expression changes in the lung tissues of plateau yaks and their environmental adaptation.
2. The goal was to enhance understanding of the biological mechanisms underlying yak adaptation to plateau environments and to provide insights for research on plateau animal conservation and adaptation.

In the text should be taken «yaks and»
GO functional analysis indicated that differentially expressed genes in the lung tissues of yaks and cattle at low altitude, as well as yaks at both high and low altitudes, are involved in processes like cell adhesion and collagen synthesis.

In the discussion authors mentions:
Studies have demonstrated that lung tissues of high-altitude indigenous animals, including Tibetan sheep, plateau pika, plateau zokor, and Himalayan dry otter, contain more elastin fibers (Krishnan et al., 2020; West et al., 2021)
However, the article (Krishnan et al., 2020) dedicated to: modifications in developmental cardiopulmonary adaptations to chronic hypoxia using a murine model of simulated high-altitude exposure.

Reviewer 2 ·

Basic reporting

The revised manuscript addresses the main concerns well. The introduction now includes relevant citations, and the connection between collagen expression and altitude acclimatization has been clarified, with references like Wiener et al., 2021, and Schreier et al., 2013, adding credibility. The hypothesis is also clearer, so this part seems resolved.

Regarding the small sample size, the authors explained why they couldn't increase it due to logistical challenges but acknowledged the issue. However, this limitation still needs to be clearly mentioned in the discussion.

The lung tissue collection procedure has been clarified with details that samples were taken within 30 minutes of death, which seems sufficient. As for the histological staining, they provided details on maintaining consistency across samples, such as using the same concentration, time, and temperature, which adds transparency to the methods.

In terms of data presentation, the authors addressed the concerns by including specific results in supplementary materials and making figures clearer. The superficial discussion of biological significance is also improved, with the addition of more references and a deeper analysis of how collagen expression relates to high-altitude adaptation.

The discussion now more clearly links the results to the initial hypothesis about collagen’s role in high-altitude adaptation, and comparisons with other studies, particularly on animals in hypoxic environments, have been expanded. However, there are still some speculative elements in the discussion that should be toned down to avoid overinterpretation of the results. These sections could benefit from more cautious wording to reflect the uncertainty where the data doesn’t fully support certain conclusions.

Overall, most issues seem well addressed, but it’s important to ensure the limitations related to sample size and any speculative interpretations are clearly acknowledged

Experimental design

The issue of small sample size persists, and although the authors explained their constraints, it’s important that this limitation is clearly addressed in the discussion. Their approach to controlling for confounding variables like species and altitude through specific group design is reasonable, though the small sample size still limits the robustness. The choice of altitudes has been adequately justified by referencing the genetic background and distribution ranges of the animals. The variability in the animal sacrifice method has been clarified, and now includes a detailed description of the timeframe for tissue collection, which seems to have resolved this concern.

Ethical compliance has also been clearly detailed, adhering to both international and Chinese standards. The authors provided more specifics on the histological methods, including controls for consistency in staining, which has improved the methods section significantly.

The use of a single reference gene (GAPDH) is acceptable, given its stability in hypoxic conditions as referenced by previous studies, but using multiple reference genes would have strengthened the study. For the RNA sequencing depth and quality control, they have provided a thorough explanation, ensuring that the analysis was adequately performed. The issue of handling multiple comparisons has also been addressed, with details of statistical corrections during RNA sequencing.

Validity of the findings

The validity issues have largely been addressed, with the authors emphasizing the preliminary nature of their findings and acknowledging that further analysis is needed in future studies .

Additional comments

The English language has been reviewed and improved, making the manuscript much clearer. The contradictions in the discussion about collagen and elastin have been resolved, with the authors providing a better explanation of their roles in lung function, supported by relevant literature. While they have justified their interpretations, some parts of the discussion still feel a bit speculative, and these areas would benefit from a more cautious tone to avoid overinterpretation.

The integration with broader literature has been handled well, with additional references adding depth to the discussion. As for alternative explanations, the authors have acknowledged that factors like diet and physical activity could also influence collagen expression, though their focus remains on altitude and species. This is acceptable, but they could explore these other factors in future studies.

---

## Round 0.3 · accepted · Accept

I congratulate you on the acceptance of this article for publication. I hope that this article will be of interest to many specialists in this field of research and will become a significant step in a deeper understanding of human and animal physiology.